# The role of contributing factors, triggers, and prodromal symptoms in the etiological classification of out-of-hospital cardiac arrest; A scoping review

Sedigheh Shaeri[1,2]*, Julie Considine[3,4], Katie N. Dainty[1,5], Theresa Mariero Olasveengen[6], Laurie J. Morrison[1,7,8]*

1 Institute of Health Policy, Management, and Evaluation, Dalla Lana School of Public Health, University of Toronto, Toronto, Canada, 2 Centre for Research and Quality, SickKids Hospital, Toronto, Canada, 3 School of Nursing and Midwifery and Centre for Quality and Patient Safety Research in the Institute for Health Transformation, Deakin University, Geelong, Australia, 4 Centre for Quality and Patient Safety Research - Eastern Health, Box Hill, Australia, 5 Office of Research & Innovation, North York General hospital, Toronto, Canada, 6 Department of Anesthesia and Intensive Care, Oslo University Hospital, and Institute of Clinical Medicine, University of Oslo, Oslo, Norway, 7 Department of Emergency Services, Sunnybrook Health Sciences Centre, Toronto, Canada, 8 The Division of Emergency Medicine, Department of Medicine, University of Toronto, Toronto, Canada

☉ These authors contributed equally to the work.
* sedigheh.shaeri@alumni.utoronto.ca (SS); laurie.morrison@sunnybrook.ca (LJM)

## Abstract

### Background

Current Utstein etiological classifications for out-of-hospital cardiac arrest (OHCA) are heterogenous and inaccurate when compared with robust sources. This heterogeneity may influence reporting incidence and outcomes and patient enrollment in observational studies and clinical trials. Circumstance-related factors may contribute to cardiac arrest; however, the role of these factors in improving the etiological classification of OHCA is unknown.

### Objective

This scoping review was proposed to explore current evidence to identify the role of contributing factors, triggers, and prodromal symptoms of out-of-hospital cardiac arrest in the reported etiology of cardiac arrest based on emergency medical services data, medical records, or autopsy reports.

### Method

We searched Medline, Embase, and EMB review-Cochrane databases from 1946 to 2024. Studies were selected if the included population was adults with OHCA for whom the initial etiology was assigned, and any contributing factors, triggers, or

**Data availability statement:** All relevant data are within the manuscript and its Supporting information files. This is a scoping review to map current published articles. No meta-analysis has been conducted. Minimal data set of this scoping review, including a summary of included studies is available within the manuscript and S3-6 Appendices with relevant citations. The result of descriptive analysis (number and percentage) is available in Table 2. No mean and standard deviation have been measured. There is no graph in this submission. No points have been extracted from images for analysis.

**Funding:** The primary author received 2019-2020 Graduate student funding from Institute of Health Policy, Management, and Evaluation and Graduate Student Endowment Funding from faculty of medicine, University of Toronto. The funders had no role in study design, data collection and analysis, decision to publish, or preparation of this manuscript.

**Competing interests:** The authors have declared that no competing interests exist.

prodromal symptoms of OHCA were reported. A descriptive review of the included studies was conducted.

## Result

The search yielded 24,833 citations. Seventy studies met the inclusion criteria. These studies were published predominantly in Europe and Asia between 2010 and 2024 and classified as contributing factors (n = 24), exercise (n = 13), environmental triggers (n = 24), and prodromal symptoms (n = 9). The etiology of cardiac arrest initially assigned to cardiac or obvious non-cardiac classification may be precipitated by seizures (n = 8), trauma (n = 7), alcohol or drug intoxication (n = 6), Covid-19 infection (n = 5), myocardial infarction (n = 4), suicide (n = 4), antipsychotic medications (n = 4), and illicit drug use (n = 3). Exercise and environmental factors (e.g., particulate matter (PM) 2.5µ and ambient temperature) may trigger cardiac arrest predominantly due to cardiac etiologies. Based on EMS data, approximately 50% of patients with OHCA experienced symptoms prior to cardiac arrest which suggested cardiac and non-cardiac etiologies.

## Conclusion

Many circumstance-related factors may directly or indirectly contribute to cardiac arrest etiology classification. Listing these factors in the reporting template may help prehospital personnel and data abstractors gather enough information to identify more accurately the etiology of OHCA.

## Introduction

The annual incidence of out-of-hospital cardiac arrest (OHCA) is about 56–100 per 100,000 people globally attributed to presumed cardiac or medical etiologies (71–90%) [1–3]. Different etiologies may cause cardiac arrest, but determining the etiology of cardiac arrest is sometimes challenging in prehospital settings. The Utstein reporting template was published to standardize OHCA-related data including etiology across all data registries [4]. Based on the 2004 Utstein template, a cardiac etiology was presumed if there was no obvious evidence of other etiologies [4]. There is a considerable disparity in published reporting of etiology of cardiac arrest documented in EMS data using the Utstein template in comparison with etiologies recorded in medical charts or autopsy reports [5]. For patients with unsuccessful resuscitation, the etiology of OHCA is assigned based on EMS data and may be over-classified within the presumed cardiac category when compared to the death certificate in as many as 40% of cases [6,7].

The Utstein etiological classification was updated in 2015, and "presumed cardiac" and "obvious non-cardiac" were replaced with "medical" and "non-medical" [8]. This updated classification is also etiologically heterogeneous. Studies demonstrated that reported outcomes analyzed based on each etiology may be varied from reported

outcomes based on lumping all cases into the 2004 or 2015 Utstein etiological classifications [9,10]. Defining etiologically homogeneous patient cohorts for data analysis may contribute to better comparability across studies and more refined recruitment into clinical trials.

Recent studies have suggested that some circumstance-related factors might potentially contribute to cardiac arrest [11–14]. Other studies highlighted that more than 30% of cardiac death due to drowning might be precipitated by a pre-existing medical condition, including alcohol intoxication, cardiac pathology, and psychotropic medication use [15,16]. In addition, previous systematic reviews suggested daily emotional stress, physical activity, cold or hot ambient temperature, and environmental stress might trigger cardiac arrest [17–19], but the association of these factors with the etiology of OHCA has not been systematically investigated. Although exercise can improve cardiovascular circulation and significantly prevent cardiovascular disease, recent studies have demonstrated that exercise might trigger cardiac arrest [20,21]. The reporting of circumstance-related factors is lacking in the current Utstein template. Identifying these factors may enable prehospital personnel and data abstractors to identify the more likely etiology of OHCA.

The aim of this scoping review is to explore current published articles to identify all circumstance-related factors of OHCA, including contributing factors, triggers, and prodromal symptoms that might refine the etiological classification of OHCA. A scoping review was considered the best method to map all relevant evidence given the anticipated diversity in methodology and limited number of studies [22].

## Method

Arksey and O'Malley's methodological steps with the refinements proposed by Levac were followed to develop the protocol and conduct this scoping review [22,23]. The international database of prospectively registered systematic reviews in health and social care (PROSPERO), Medline, google scholar, and open science framework were checked to confirm that no systematic, scoping, or narrative reviews on a similar topic have been published.

This review was reported in accordance with the Preferred Reporting Items for Systematic Reviews and Meta-Analyses extension for Scoping Reviews (PRISMA-ScR) (S1 Appendix) [24]. The reviewer followed the written protocol and applied data screening steps and a standard data abstraction form available on https://www.covidence.org [25]. The ethics approval was not required as all sources of data informing this scoping review are publicly available. The protocol was registered (OSF.IO/K5ZDP) with Open Science Framework (OSF) at https://doi.org/10.17605/OSF.IO/K5ZDP [26]. Our scoping review consisted of the following steps:

### Identifying research question

The research question was: What contributing factors, triggers, and prodromal symptoms aid prehospital personnel in their effort to assign more accurately the etiology of cardiac arrest?

### Search strategy

The search strategy was developed by an experienced information specialist (DL) and is available in the supplemental material. Medline, Excerpta Medica database (Embase), and evidence-based medicine (EBM) review-Cochrane electronic databases were searched on 31 January 2021 and updated on 28 June 2024 by using a combination of indexes and Mesh terms. (Full search strategy and mesh terms: S2 Appendix)

### Eligibility criteria to select studies

Table 1 explains the inclusion criteria for selecting articles. The **population** was limited to adult patients (as defined in each paper) who had experienced OHCA and were treated by EMS personnel and for whom the initial diagnosis was assigned. The **outcome** of interest was to identify any reported contributing factors, triggers, and prodromal symptoms that might refine the etiological assignment of cardiac arrest.

**Table 1. Inclusion criteria for selecting studies.**

| | |
|---|---|
| **Population** | Adult patients (as defined in each paper) who had experienced OHCA and were treated by EMS and for whom the initial diagnosis was assigned. |
| **Outcome of interest** | Any reported contributing factors, triggers, and prodromal symptoms that might refine the etiological assignment of cardiac arrest |
| **Study designs** | Limited to randomized control trials (RCT), experimental, observational studies, and those that reported initial or final diagnosis of OHCA. Guidelines, editorial reviews, conference abstracts, and commentaries were excluded. |
| **Year and Language** | Literature published in **English** between **1946 and 2024**. |

**OHCA**: out-of-hospital cardiac arrest

## Source of evidence selection

All citations were uploaded into the Covidence website for screening (www.covidence.org) [25]. All duplicates were excluded. First, the primary author (SS) reviewed all titles and abstracts against inclusion and exclusion criteria. For initial screening, limited inclusion criteria were employed in order to have broader inclusion and minimize potential selection bias. After initial screening, all potential eligible full texts were retrieved and further reviewed by the primary author (SS) against the same eligibility criteria. Additional citations were found through hand searching of reference lists of included studies following the initial review. Whenever there was uncertainty about a potential eligible study, the senior author (LJM) provided an additional review to verify selected full texts, and final decisions were achieved by discussion and consensus. The primary author reviewed the final selected articles multiple times to ensure the accurate selection.

## Data extraction and charting the data

The primary author (SS) followed the protocol and the JBI (Joanna Briggs Institute) methodological guideline [27] to extract the following data:1- year and geographical origin of publication, 2- design of study, 3- number of included patients, 4-source of initial and determined etiologies of OHCA if reported, 5- detailed initial and determined etiologies of OHCA if reported, 6-contributing factors, triggers, and prodromal symptoms, and 7- other related information. The senior Author (LJM) reviewed the abstracted data to verify the accuracy of data.

## Collating and synthesizing data

All included articles were grouped based on contributing factors, environmental factors, exercise- induced OHCA, and prodromal symptoms in order to present explicitly the association of these factors with the etiology of OHCA. Basic descriptive analyses (e.g., numbers and percentages) were computed to present the prevalence of each characteristic of included studies. A narrative description was provided to explain the result of this review. No formal critical appraisal of certainty was undertaken in accordance with the defined methodology of scoping reviews [27–29]. Tables were plotted when they were necessary to present data.

## Results

### Study selection

The search yielded 24,833 (103 citations from EBM review-Cochrane (2005–2024), 4,786 citations from Embase (1947–2024), 19,942 citations from Medline (1946–2024), and 2 from hand searching of bibliographies) articles to screen. After excluding 6,156 duplicates, 18,677 citations were imported into the Covidence to review for titles and abstracts, and 1,108 full texts were assessed according to inclusion and exclusion criteria. Fig 1 presents the process of selecting articles.

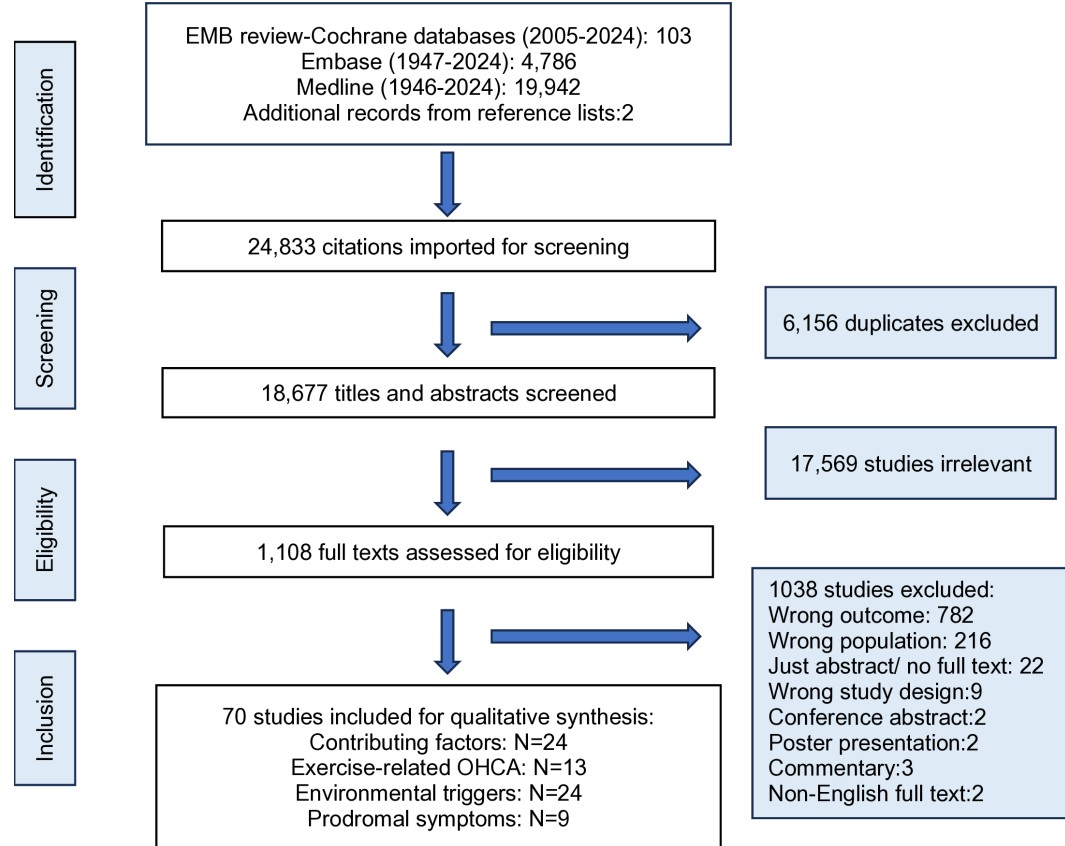

**Fig 1. PRISMA flowchart. EBM**: Evidence-based medicine. **Embase**: Excerpta Medica Database. **OHCA**: Out-of-hospital cardiac arrest.

## Studies characteristics

In total, 70 studies were included for data abstraction. The summary of included studies was presented in S3–S6 Appendices. The included studies were grouped to contributing factors (n = 24) [7,13,30–51], exercise- related OHCA (n = 13) [20,52–63], environmental factors (n = 24) [64–87], and prodromal symptoms (n = 9) [88–96]. Table 2 describes the general descriptive characteristics of included studies by circumstance-related factors of OHCA.

## Contributing factors

In total, twenty-four studies discussed the contributing factors of OHCA, including Covid-19 (n = 5) [30–33,48], antipsychotic/ antidepressant medications (n = 3) [13,34,35], drug overdose (n = 3) [7,13,41], epilepsy (n = 5) [42–45,49], and other contributing factors (n = 9) [36–40,46,47,50,51] (S3 Appendix). Table 3 summarizes the initial presumed etiology of OHCA and potential contributing factors found in included studies.

 **Covid-19 infection.** Table 4 reports the data from five studies that compared the epidemiological characteristics of OHCA during Covid-19 outbreak between Feb-April 2020 with the same period in 2019 [30–33,48]. This is an example of a confirmed Covid-19 infection as a contributing factor to etiological classification of OHCA. The incidence of OHCA due to respiratory diseases or asphyxia was reported to be higher following Covid-19 outbreak compared with the pre-covid period (54% vs 9.9%;OR:11.11 (6.67–16.1) p < 0.001) [30] (20% vs 16%) [32] (10.5% vs 3.9%) [48], and the incidence of medical etiologies was higher during pandemic than pre-covid period (30% vs 18.2%) [31] (94.9% vs 80%) [33]. The

**Table 2. Descriptive characteristics of included studies by circumstance-related factors of OHCA.**

| | Contributing factors; n | Exercise related-OHCA; n | Environmental factors; n | Prodromal symptoms; n | Total; n(%) |
|---|---|---|---|---|---|
| **Number of articles** | 24 | 13 | 24 | 9 | 70 (100) |
| **Regional origin of included studies:** | | | | | |
| Europe [20,30–33,35,37,40,43-49,51,54,55,59–63, 67,72,74,84,85,87,92,93] | 15 | 8 | 5 | 3 | 31(44) |
| Asia [36,52,53,57,58,64–66,69–71,73,75,77,78,86,88,89,95,96] | 1 | 4 | 11 | 3 | 19 (27) |
| North America [7,13,34,38,41,42,50,56,68,79,82,94,96] | 7 | 1 | 3 | 2 | 13 (19) |
| Australia [25,55,59,60,62,66] | 1 | | 4 | 2 | 7 (10) |
| **Type of studies design:** | | | | | |
| Cohort studies[7,13,20,30,35,38–41,46,47,50,52,57,77,88,95] | 11 | 3 | 1 | 2 | 17 (24) |
| Observational study[31–34,36,37,42–45,48,49,51,53–56, 58–63,66,73,74,78,84,86,89,92–94,96,97] | 13 | 10 | 6 | 6 | 35 (50) |
| Case-crossover [65,67,69,71,72,76,79-82,85,87,90] | | | 12 | 1 | 13 (19) |
| Case-crossover and time series [64,68,70,75,83] | | | 5 | | 5 (7) |

OHCA: out-of-hospital cardiac arrest.

incidence of asphyxia or respiratory disease as the etiology of cardiac arrest was higher among patients with positive Covid-19 test than patients with negative Covid-19 test (25% vs 11.3%) [48] (56.9% vs 12%) [32].

**Drug overdose (OD).** Three cohort studies compared the etiology of non-survivors of OHCA assigned to presumed cardiac (Utstein 2004) with autopsy reports [7,13,41]. Two studies reported occult drug overdose and positive toxicology test were the primary etiology of OHCA. Tseng et.al identified occult drug overdose in 13.5% of cases who were initially classified as presumed cardiac (Utstein 2004) etiologies (n = 71/525). In these cases, 61% had a fatal toxicology level of opioid (43/71) [7]. Another study reported that 17% (133 out of 756) of OHCA assigned to presumed cardiac (Utstein 2004) etiologies were diagnosed with drug overdose through autopsy examination and suggested an algorithm for assisting paramedics to suspect drug overdose as the more correct etiological classification in similar cases [41]. One study identified 31% of patients with OHCA of presumed cardiac etiologies had positive toxicology tests with high blood level of one or more than one drug, including cocaine (43%), ethanol (6%), and opioid (38%) [13].(Table 4)

**Epilepsy (Seizure).** Five studies investigated the association of seizure and anti-seizure medications with presumed cardiac OHCA [42–45,49], and three studies identified seizure as a contributing factor of OHCA due to drowning [36,37,50]. One case-control study reported that drug poisoning (including, tricyclic antidepressant (n = 5), neuroleptic (n = 2), cocaine and amphetamine (n = 1)) ((28.6% vs 9%;p = 0.002), acute alcohol intoxication (18.4% vs 13.2%; p = 0.3), traumatic brain injury (6.1% vs 3.4%;p = 0.4), and metabolic disorders (12.2% vs 11.5% p = 0.8) were contributing factors of OHCA with etiology of epilepsy when compared to epilepsy patients who did not arrest [44]. Two case-control studies identified that risk of OHCA is higher in patients with history of active seizure (HR = 1.76; 95% CI 1.64–1.88) [49] (OR = 2.9; 95% CI = 1.1−8;p = 0.034) [43] and anti-epileptic medication use (clonazepam (HR = 1.88; 95% CI 1.45–2.44) [49] and pregabalin (HR = 1.33; 95% CI = 1.05–1.69)) [49]. (Table 4)

In contrast, two studies suggested that assigning epilepsy as the etiology of cardiac arrest may be masking an underlying cardiac etiology [42,45]. Patients with a history of seizure are more likely to show seizure-like activity before cardiac arrest (34% vs 10%;p = 0.01) [42]; however, 50% of patients with cardiac arrest and history of seizure had obstructive CAD following autopsy examination [42]. Clinical cardiac disease was more common in epileptics when compared to patients without a history of epilepsy (clinical heart disease: 50% vs 15%; OR = 6.87 (1.29–36.5), including acute myocardial infarction (44% vs 57%; OR = 1.19 (0.3–3.7)) [45].

**Table 3. Summary of contributing factors of out-of-hospital cardiac arrest (OHCA) identified in included studies.**

| Author's name | OHCA due to presumed cardiac etiologies or medical etiologies | | | | | OHCA due to non-medical etiology@ | | Contributing factors | | | | | | | | | | | | | | | | | |
|---|---|---|---|---|---|---|---|---|---|---|---|---|---|---|---|---|---|---|---|---|---|---|---|---|---|
| | SCD following unsuccessful resuscitation& | Presumed cardiac or medical etiologies | Neurological etiologies | Respiratory disease | Hypothermia | Convulsion | Drowning | Covid-19 infection | Antipsychotic medications' | Alcohol intoxication | Drug intoxication | Trauma/MCA | Seizure'' | Coronary artery sclerosis | Hypothermia | MI | Hypoglycemia | Stroke/neurological event | Illicit substance | Suicide | Drug overdose | QT prolongation drugs | Metabolic disorder | Vascular disease | End stage disease/malignancy |
| Hubert [30] | | | | × | | | | × | | | | | | | | | | | | | | | | | |
| Fothergill [31] | | | | × | | | | × | | | | | | | | | | | | | | | | | |
| Baert [32] | | × | | | | | | × | | | | | | | | | | | | | | | | | |
| Baldi [33] | | × | | | | | | × | | | | | | | | | | | | | | | | | |
| Sultanian [48] | | | | × | | | | × | | | | | | | | | | | | | | | | | |
| Teodorescu [34] | × | | | | | | | | × | | | | | | | | | | | | | | | | |
| Kauppila [35] | | × | | | | | | | × | | | | | | | | | | | | | | | | |
| Allan [13] | | × | | | | | | | × | | | | | | | | | | × | | | | | | |
| Youn [36] | | | | | | | × | | | × | × | × | × | | | | | × | | | × | | | | |
| Claesson [37] | | | | | | | × | | | | | × | × | × | × | × | × | × | | | | | | | |
| Reynold [38] | | | | | | | × | | | | | | | | | | | | × | | | | | | |
| Dyson [39] | | | | | | | × | | | | | × | | | | | | | | × | | | | | |
| Grmec [40] | | | | | | | × | | | × | | | | | | | | | | × | | | | | |
| Rodriguez [41] | × | | | | | | | | | | | | | | | | | | | | × | | | | |
| Tseng [7] | × | | | | | | | | | | | | | | | | | | | | × | | | | |
| Stecker [42] | | × | | | | | | | | | | | × | | | | | | | | | | | | |
| Bardi [43] | | × | | | | | | | | | | | × | | | | | | | | | | | | |
| Legriel [44] | | | | | | × | | | × | × | × | × | | | | | | | × | | | | × | × | |
| Lamberts [45] | | × | | | | | | | | | | | × | | | × | | | | | | × | | | |
| Eroglu [49] | | × | | | | | | | | | | | × | | | | | | | | | | | | |

*(Continued)*

| Author's name | OHCA due to presumed cardiac etiologies or medical etiologies | | | | | | OHCA due to non-medical etiology@ | Contributing factors | | | | | | | | | | | | | | | | | |
|---|---|---|---|---|---|---|---|---|---|---|---|---|---|---|---|---|---|---|---|---|---|---|---|---|---|
| | SCD following unsuccessful resuscitation& | Presumed cardiac etiologies or medical etiologies | Neurological etiologies | Respiratory disease | Hypothermia | Convulsion | Drowning | Covid-19 infection | Antipsychotic medications* | Alcohol intoxication | Drug intoxication | Trauma/MCA | Seizure** | Coronary artery sclerosis | Hypothermia | MI | Hypoglycemia | Stroke/neurological event | Illicit substance | Suicide | Drug overdose | QT prolongation drugs | Metabolic disorder | Vascular disease | End stage disease/malignancy |
| Legriel [46] | | | × | | | | | | | × | × | | × | | | | | | | | | | | | |
| Schober [47] | | | | | × | | | | | | × | × | | | | | | | | | | | × | | × |
| Ryan [50] | | | | | | | × | | | × | × | × | × | | | × | | | | × | | | | | |
| Reizine [51] | | | | | | | × | | | × | × | × | | | | × | | × | | × | | | | | |

**OHCA**: Out-of-hospital cardiac arrest. **MCA**: Motor car accident. **MI**: Myocardial infarction. **SCD**: Sudden cardiac death.

@Non-medical or obvious non-cardiac etiologies.

&Non-survivors of OHCA due to presumed cardiac etiology.

*Antipsychotic medications include antipsychotic, antidepressant, and psychotropic medications.

**Seizure includes all cases with the history of seizure, epilepsy, or antiepileptic medication use.

**Table 4. Summary of reported results from each included studies informed the association of contributing factors with out-of-hospital cardiac arrest (OHCA) etiologies.**

| Author | Initial etiological classification of OHCA;n | Identified contributing factors/Triggers | Etiologies associated with contributing factors of OHCA: n(%) |
|---|---|---|---|
| **Covid-19 infection** | | | |
| Hubert [30] | Presumed cardiac etiologies (Utstein 2004) (n = 670) | Covid-19 infection | **Covid-19 positive (n = 146)**<br>Respiratory etiology:54.1% (OR:11.11 (6.67-16.67), p < 0.001)<br>Cardiac etiologies: 34.9% (OR: 0.20 (0.13-0.30), p < 0.001)<br>Other etiologies:11% (OR: 0.6(0.31–1.06) p = 0.73) |
| | | | **Covid-19 negative (n = 524)**<br>Respiratory etiologies of OHCA:9.9%<br>Cardiac etiologies: 72.9%<br>Other etiologies:17.2% |
| Fothergill [31] | Presumed cardiac etiologies (Utstein 2004) (n = 4,869) | Covid-19 infection | **Covid-19 period(n = 3122)**<br>Presumed cardiac: 66.7%; p < 0.001<br>Medical:30.8%<br>Trauma:2.5%<br>**Covid-19 positive (n = 2356)**<br>Presumed Cardiac:55%<br>Medical: 44.8%<br>Trauma:0%<br>**Covid-19 negative (n = 766)**<br>Presumed Cardiac:72.8% p < 0.001<br>Medical: 23.5%<br>Trauma:3.8% |
| | | | **Pre-Covid-19 period (n = 1724)**<br>Presumed cardiac:76.4%<br>Medical:18.2%<br>Trauma: 5.4% |
| Baert [32] | All medical OHCA (Utstein 2015) (n = 2,625) | Covid-19 infection | **Covid-19 period:**<br>Cardiac: 726(72.2%)<br>Respiratory: 210(20.9%)<br>Other medical cause: 50(5%)<br>**Covid-19 positive (n = 197):**<br>Cardiac: 67(34%)<br>Respiratory:112(56.9%)<br>Other medical cause:16(8.1%)<br>**Covid-19 negative (n = 808) (p < 0.001)**<br>Cardiac: 659(81.6%)<br>Respiratory: 98 (12.1%)<br>Other medical causes:34(4.2%) |
| | | | **Pre-Covid-19 period:**<br>Cardiac: 1275 (78.7%)<br>Respiratory:260 (16%)<br>Other medical cause: 60(3.7%) |
| Baldi [33] | Medical OHCA (Utstein 2015) (n = 811) | Covid-19 infection | **Covid-19 period:**<br>Medical etiologies: 94.9%<br>**Covid-19 positive:**<br>Medical etiology: 100%<br>**Covid-19 negative:**<br>Medical etiologies: 93% |
| | | | **Non-Covid-19 period:**<br>Medical etiologies: 80.2% |

*(Continued)*

| Author | Initial etiological classifica-tion of OHCA;n | Identified contributing factors/Triggers | Etiologies associated with contributing factors of OHCA: n(%) |
|---|---|---|---|
| Sultanian [48] | All etiologies (Utstein 2015) (n = 1,946) | Covid-19 infection | **Covid-19 period:**<br>Medical etiologies: 80%<br>Asphyxia: 10.5%<br>Trauma:3.3%<br>**Covid-19 positive**<br>Medical: 73.5%<br>Asphyxia: 25%<br>Trauma: 1.5%<br>**Covid-19- negative**<br>Medical etiologies: 81.3%<br>Asphyxia: 11.3%<br>Trauma: 2.5%<br><br>**Non-Covid-19 period:**<br>Medical cause:90%<br>Asphyxia:3.9%<br>Trauma: 2.2% |
| **Drug overdose** | | | |
| Rodriguez [41] | Non-survivors of OHCA pre-sumed cardiac (Utstein 2004) (n = 767) | Drug overdose | Opioid overdose (n = 79)<br>Non- opioid overdose (n = 54) |
| Tseng [7] | Non-survivors of OHCA (presumed cardiac etiologies) (n = 525) | Drug overdose | Occult overdose: 71(13.5%)<br>Lethal toxicological levels of opiates:61% (43/71)& |
| Allen,K [13] | Non-survivors of OHCA (presumed cardiac etiologies) (Utstein 2004) (n = 608) | Illicit medications | Cocaine/benzoylecgonine: 16<br>Alcohol/opioid:12 |
| **Seizure** | | | |
| Legriel [44] | CSE-related OHCA(N = 49) | 1-Potential drug poisoning<br>2-Other medical conditions | **Drug poisoning:**14 (29%) (OR:3.69(1.7–7.8) p = 0.0007) (including, psychotropic drugs, tricyclic antidepressant, rec-reational drug (cocaine and amphetamine), and prescribed medication)<br>**Acute alcohol intoxication/ withdrawal**: 9 (18.4%)<br>**Vascular disease**:7(14.3%)<br>**Metabolic disorder**: 6(12.2%)<br>**Undetermined**: 5(10.2%) |
| | Seizure without CA(N = 235) | | **Drug poisoning:**23(9.8%) p = 0.002<br>**Acute alcohol intoxication/ withdrawal**: 31(13.2%), p = 0.37<br>**Vascular disease**:44(18.7%) p = 0.54<br>**Metabolic disorder**: 27(11.5%) p = 0.81<br>**Undetermined**: 27(11.5%) p = 1 |
| Bardie [43] | Epilepsy history with CA (N = 1,019) (Presumed cardiac etiology) (Utstein 2004) | Seizure | Cardiac causes: OR: 2.9 (1.1–8) p = 0.034 |
| | Epilepsy history without CA (N = 2,834) | | |
| Stecker [42] | Presumed cardiac etiologies of OHCA patients with history of epilepsy(N = 106) (Utstein 2004) | 1- Seizure<br>2- Antiepileptic medication<br>3- Other precipitating factors | Seizure-like activity before CA:34%^<br>Antiepileptic medication:79% |
| | Presumed cardiac OHCA without history of epilepsy (N = 2,311) | | 1- Seizure-like activity before CA = 10%; p = 0.001<br>2- Antiepileptic medication:16%; p < 0.0001 |

*(Continued)*

| Author | Initial etiological classification of OHCA;n | Identified contributing factors/Triggers | Etiologies associated with contributing factors of OHCA: n(%) |
|---|---|---|---|
| Lambert [45] | Presumed cardiac OHCA cases with history of epilepsy (Utstein 2004) (N = 18) | 1-Cardiac disease 2-Medications | 1-Clinically relevant heart disease: 50% (OR:6.87 (1.29–36.5)) 2-Acute myocardial infarction:44% (OR:1.19 (0.3–3.7)) 3-Other potential contributing factor: QT-prolongation drugs: 28% Depolarization -blocking drugs: 56% |
| | OHCA without history of epilepsy (N = 470) | | 1-Clinically relevant heart disease: 15%p = 0.005 2-Acute myocardial infarction: 57% Other potential contributing factor: QT-prolongation drugs: 24% (OR:1.20 (0.37–3.92) Depolarization -blocking drugs: 50% (OR:1.25 (0.43–3.61)) |
| Eroglu [49] | OHCA with presumed cardiac etiologies (N = 35,195) | Seizure or anti-epileptic medications | Epilepsy associated with increased risk of OHCA: HR: 1.76 (1.64–1.88) Pregabalin: HR:1.33 (95% CI = 1.05–1.69) Clonazepam: HR:1.88 (95% CI 1.45–2.44) |
| | Non OHCA cases (N = 351,950) | | |
| **Antipsychotic medications** | | | |
| Teodorescu [34] | OHCA with presumed cardiac (Utstein 2004) etiologies with PEA as the initial rhythm (N = 309) | Antipsychotic or Antidepressant medications | Antipsychotics: 13.6% p < 0.0001 (OR:2.40(1.26–4.53; p:0.007)) Antidepressants: 30.7% p = 0.004 (OR:1.13(0.76–1.68)) Digoxin: 5.8% p:0.001 (OR:0.28 (0.14–0.53; p < 0.0001)) |
| | OHCA with presumed cardiac with VT/VF as the initial rhythm (N = 509) | | Antipsychotics: 4.1% Antidepressants: 21.6% Digoxin: 13% |
| Kauppila [35] | OHCA with presumed cardiac etiologies and non-shockable rhythm (N = 99) | 1-Antipsychotic medications 2-Antidepressant medications | Psychotropic medication: 26(26%); AOR:3.18(1.4–7.23) p = 0.006 Benzodiazepine: 12(12%); OR:3(0.96–9.35) p = 0.058 Antidepressant: 14(14%); OR:2.11(0.73–6.15) p = 0.17 Antipsychotic: 12(12%); OR:4.27 (1.28–14.2) p = 0.018 Multiple psychotropic medications:10(10%) |
| | OHCA with shockable initial rhythm (N = 123) | | Psychotropic medication: 10(8%) p = 0.003 Benzodiazepine: 5(4%) p = 0.009 Antidepressant: 6(5%) p = 0.031 Antipsychotic: 5(4%) p = 0.016 Multiple psychotropic medications:5(4%) |
| Allen [13] | OHCA of no obvious etiology (N = 608) | 1-Psychotropic medications 2-History of psychiatric diseases | Identified medications through toxicology test: SSRI:18 Benzodiazapine:16 Dopamine antagonist:13 Antidepressant/antipsychotics:11 |
| **Other identified contributing factors** | | | |
| Youn [36] | OHCA due to drowning (N = 131) | 1- Intoxication 2-Trauma 3-Seizure | Unknown:103(78.6%) Alcohol intoxication: 18(13.7%) Drug intoxication:2(1.5%) Traumatic injury:4(3.1) Seizure:2 |

*(Continued)*

**Table 4.** (Continued)

| Author | Initial etiological classification of OHCA;n | Identified contributing factors/Triggers | Etiologies associated with contributing factors of OHCA: n(%) |
|---|---|---|---|
| Claesson [37] | OHCA due to drowning (N = 2,438) | 1-Cardiac disease 2-Trauma 3-Neurological disease 4- Seizure 5-Other medical and non-medical conditions | Coronary artery sclerosis: 137 (5.3%)* Hypothermia:134 (5.2%) Trauma:86(3.3%), Myocardial infarction:50 (1.9%), Other cardiac conditions:49 (1.8%). Epilepsy:44 (1.7%) Stroke:14 (0.5%) Hypoglycaemia:9 (0.3%) Other causes:39 (1.5%) |
| Reynold [38] | OHCA due to drowning(N = 407) | Illicit substances | Precipitating illicit substances: 45(11.1%) |
| Dyson [39] | OHCA due to drowning (N = 336) | 1-Trauma 2-Suicide | **Primary suspected cause of drowning**: MVA:3(0.9%) Suicide:3(0.9%) |
| Grmec [40] | OHCA due to drowning (N = 32) | 1-Alcohol intoxication 2-Suicide | Alcohol intoxication: 38% Suicide: 69% |
|  | OHCA due to presumed cardiac (Utstein 2004) (N = 528) |  | Alcohol intoxication: 12% Suicide:0.4% |
| Ryan [50] | OHCA due to drowning (n = 1,767) | 1-Cardiac disease 2-Trauma 3-Neurological disease 4-Seizure 5-Suicide 6-Intoxication 7-Other medical and non-medical conditions | Unknown: 52.3% Alcohol/ drug use:6.5% Hyperventilation/Breath holding: 0.3% Seizure:1.9% Suspected cardiac causes:1.9% Suicide:0.7% Traumatic injury:1.1% Other: 7.3% |
| Reizine [51] | Drowning -related OHCA (Utstein 2015) (n = 103) | 1-Cardiac disease 2-Intoxication 3-Neurological disease 4-Suicide 5-Trauma | Presumed cardiac:12.6% Drug or alcohol intoxication:17.5% Presumed neurologic: 7.8% Suicide attempt:9% Accident: 53.4% |
| Legriel [46] | OHCA due to neurological etiologies (N = 247) | 1-Poisoning 2-Trauma 3-Neurological disease 4- Seizure 5-Other medical and non-medical conditions | Neurovascular etiologies:116(47%) Alcohol/ drug poisoning: 55(22.3%) Isolated traumatic brain injury (35(14.2%) Seizure: 31(12.6%) Miscellaneous:10 (4.1%) |
| Schober [47] | OHCA due to accidental hypothermic (N = 18) | 1-Intoxication 2-Trauma 3-Other medical and non-medical conditions | Drug intoxication:12(67%) Trauma:1 Metabolic disorder:3(17%) End stage disease:1 Unknown:1 |

**ACS**: Acute coronary syndrome. **AOR**: Adjusted odd ratio. **CA**: Cardiac arrest. **CAD**: Coronary artery disease. **CSE**: Convulsive status epilepticus. **CI**: Confidence interval. **HR**: Hazard ratio. **PEA**: Pulseless electrical activity. **MVA**: Motor vehicle accident. **OHCA**: Out-of-hospital cardiac arrest. **OR**: Odd ratio. **SSRI**: Selective serotonin reuptake inhibitors. **VT**: Ventricular tachycardia. **VF**: Ventricular fibrillation.

&EMS documented cardiac arrest due to presumed cardiac etiologies and no evidence or suspicion of drug use at the scene

^Autopsy report is available for 28 patients with CA and epilepsy, including: 1-Normal autopsy:14(50%) 2-Obstructive CAD:10. 3-Cardiomegaly:9

*Contributing factors which resulted in drowning were found in 21% of all cases.

**Antipsychotic drugs.** Three studies investigated the association of antipsychotic medications with presumed cardiac etiologies of OHCA [13,34,35]. Two studies examined the association of antipsychotic medications with initial non-shockable rhythm, but definitive etiologies of cardiac arrest were not reported [34,35] (antipsychotic: 13.6% vs 4.1%;

 

p < 0.0001) [34] (12% vs 3%) [35], anti-depressant (30.7% vs 21.6%; p = 0.004) [34] (14% vs 4%) [35]. Specifically, both studies identified a significant association of these medications with pulseless electrical activity (PEA) as the initial rhythm (OR:2.4 (1.26–4.53); p = 0.007) [34] and (OR= 4.27 (1.28–14.2);p = 0.018) [35]. One cohort study demonstrated that toxic blood level of tricyclic antidepressant medications in 12.5% of patients with presumed cardiac OHCA following autopsy and toxicology tests [13]. (Table 4)

**Other contributing factors.** Table 4 also includes studies identifying contributing factors that may suggest the etiology of OHCA in accidental hypothermia (n = 18) [32], and direct or indirect factors contributing to neurological etiologies of OHCA [46] or drowning -induced OHCA [36–40,50,51]. Across these studies, drowning potentially occurred in some cases (with varied frequencies) secondary to contributing factors, including coronary artery disease (CAD), hypothermia, trauma, myocardial infarction (MI), epilepsy, stroke, hypoglycemia, alcohol intoxication, drug intoxication, and suicide [36–40,50,51].

## Triggers

In total, 37 studies discuss the triggers of OHCA, including exercise-related OHCA (n = 13) and environmental factors (n = 24).

**Exercise.** Table 5 summarizes the evidence across 13 studies on exercise- related OHCA [20,52–63] with varied annual incidence rate across studies: 0.01% per year [20],0.1–0.38 per 1,000,000 people [63], 0.6 per 100,000 people [54], 0.76 per 100,000 people [56],1.67 per 100,000 people [60], 2.1 per 100,000 people [59], 3.1 per 1,000,000 people [57], 2.8% per 1,000,000 people [52], and 4.7 per 1,000,000 people/year [62]. Included populations were not the same across studies. The initial etiology of OHCA was retrieved from EMS data based on 2004 Utstein template (n = 13) compared with the final diagnoses derived from medical charts or autopsy reports [20,52–63].

In total, eleven studies defined the exercise-related OHCA as a cardiac arrest which occurred during or within a time interval of 15–60 minutes after moderate to vigorous exercise [20,52–54,56–59,61–63]. One study had a slightly different definition and included any fatal OHCAs occurring before, during, or within two hours after the end of race [60]. One study described exercise intensity as light (metabolic equivalents (MET)=0–3), moderate (MET 3–6) and vigorous (MET > 6) [58]. (S4 Appendix)

In total, eight studies reported the underlying etiologies of exercise-related OHCA based on further assessment following resuscitation [20,52,55,56,60–63]. Cardiac etiologies were the predominant etiology of OHCA among patients with SR-OHCA when compared to non-exercise-related OHCA (97% vs 80%; p < 0.001) [20]. Cardiac etiologies were more common than non-cardiac etiologies in exercise-related OHCA (90% vs 5%) [52] (98% vs 2%) [61]. One study compared the underlying etiology of exercise-related OHCA in women with men [63]. (Table 5)

**Association of environmental factors with OHCA.** Table 6 summarizes the evidence from 24 studies on the association of environmental factors with OHCA, including air pollution (n = 11) [67–69,71,72,75,76,79,80,84,85], cold or hot ambient temperature (n = 10) [64–66,70,73,74,77,78,86,87], wildfire exposure (n = 2) [81,82], and thunderstorm (n = 1) [83]. (S5 Appendix)

Ten studies evaluated the association of air pollution (e.g., particular matter (PM) 2.5 or 10 μ, carbonic monoxide (CO), Ozone (O$_3$), and pollutant standard index (PSI)) with the incidence of presumed cardiac etiology (Utstein 2004) of OHCA [67–69,71,72,75,76,79,80,84]. The results varied across studies. One study reported this association with confirmed cardiac etiologies of OHCA (acute myocardial infarction (AMI) and other cardiac diseases) [85]. The risk of OHCA due to presumed cardiac (Utstein 2004) and medical (Utstein 2015) etiologies increased with 10 (μg/m$^3$) increase in PM 2.5 (3.61% (95% CI 1.29–5.9%) [76], (1.3% (95% CI 0.2–2.4%) p = 0.02) [69], (RR: 1.06(1.02–1.1) [68], (OR: 1.02 (1–1.04);p < 0.002) [84]. This increased risk was associated with different exposure time intervals (Lag 0, 1, or 2) [69,72,75,80,85]. In contrast, two studies demonstrated no significant association between PM and incidence of OHCA (RR: 0.87 (0.74–1.01) [79] (OR: 1.6% (0.1–3.1%)) [71].

**Table 5. Summary of reported results from each included studies informed the association of etiologies with exercise-related out-of-hospital cardiac arrest.**

| Author | Initial etiological classification of OHCA | Identifies triggers | Etiologies associated with exercise-related factors of OHCA; N(%) |
|---|---|---|---|
| Soholm [20] | OHCA with presumed cardiac (Utstein 2004) etiologies | Exercise-related (N = 91) | Cardiac:88 (97%)<br>ACS:53(59%)<br>Premature ACS:7(13%)<br>STEMI:26(29%) |
| | | Non-exercise-related (N = 1233) | Cardiac:981(80%) p < 0.001<br>ACS: 474(39%) p < 0.001<br>Premature ACS:44(9%) p = 0.3<br>STEMI: 273 (23%) p = 0.1 |
| Kiyohara [57] | OHCA with presumed cardiac (Utstein 2004) | Exercise-related(N = 52) | **Cardiac causes**: N = 47 (90%)<br>**Non- cardiac causes:** N = 5 (10%)<br>SAH = 2<br>Pulmonary stenosis = 1<br>Drowning = 1<br>Injury = 1 |
| Jung [53] | OHCA with presumed cardiac etiologies (Utstein 2004) | Exercise in mountain (N = 68) | Final etiologies not reported |
| | | Exercise in other places (N = 1767) | |
| Edwards [54] | OHCA of presumed cardiac etiologies (Utstein 2004) | Exercise-related (N = 100) | Final etiologies not reported |
| | | Non-exercise related (N = 6613) | |
| Viglino [55] | OHCA of presumed cardiac etiologies (Utstein 2004) | On ski slope (N = 136) | **Cardiac etiologies**: 95 (69.9%)<br>**Non-cardiac etiologies:**<br>Trauma: 33(24.3%)<br>Respiratory: 0(0)<br>Other: 8(5.9%) |
| | | Other locations(N = 12,500) | **Cardiac etiologies**: 8092(64.7%) (p: 0.22)<br>**Non-cardiac etiologies:**<br>Trauma: 1274(10.2%) (p < 0.001)<br>Respiratory: 1412 (11.3%)<br>Other:1724 (26.8%) (p < 0.001) |
| Landry [56] | OHCA of presumed cardiac (Utstein 2004) etiologies | OHCA occurred during competitive exercise(N = 12) | Ischemic:3 (25%)<br>Primary arrhythmia: 6(50%)<br>Anomalous coronaries:2(16%)<br>Hypertrophic cardiomyopathy: 1(8%) |
| | | OHCA occurred during non-competitive(N = 58) | Ischemic: 26 (44%)<br>Arrhythmia:5(8%)<br>Structural:14(24%)<br>Unknown:2(3.4%)<br>Others: 2(3.4%) |
| Kiyohara [52] | OHCA with presumed cardiac etiologies (Utstein 2004) | Exercise-related(N = 186) | Presumed cardiac = 186 (100%)<br>Final etiologies not reported |
| | | Non-exercise -related (N = 10,873) | |
| Ro [58] | OHCA of presumed cardiac etiologies (Utstein 2004) | Exercise-related: (N = 762)*<br>MET:<br>0-3: 18.1%<br>3-6: 42.1%<br>>=6: 39.8% | Underlying final etiology not reported. |
| | | Non-exercise-related: (N = 5,511)*<br>MET:<br>0-3: 92.9% (p < 0.001)<br>3-6: 5%<br>>=6: 0.6% | |

*(Continued)*

**Table 5.** (Continued)

| Author | Initial etiological classification of OHCA | Identifies triggers | Etiologies associated with exercise-related factors of OHCA; N(%) |
|---|---|---|---|
| Berdowski [59] | All OHCA with presumed cardiac etiologies (Utstein 2004) | Exercise-related (N=143) | Final etiologies not reported |
| | | Non-exercise-relate (N=2381) | |
| Bohm [61] | OHCA of presumed cardiac etiologies (Utstein 2004) | Exercise-related OHCA(N=349) | **Cardiac etiologies:(98%)**<br>**>35 years old**:<br>Unresolved:40%<br>CAD:30%<br>Considered CAD:16%<br>Myocarditis: 4%<br>Idiopathic VF:3%<br>Cardiopulmonary:2.6%<br>**<= 35 years old:**<br>Unresolved:38%<br>Premature CAD: 14%<br>Idiopathic VF=14%,<br>Myocarditis:12%,<br>Cardiomyopathy:10%<br>**Non- cardiac etiologies**:5(1.4%)<br>PE:2<br>AAD:1<br>SAH:1<br>Severe hyponatremia: 1 |
| Gerardin [60] | OHCA of presumed cardiac (Utstein 2004) | SR-OHCA(N=18)** | **Cardiac etiologies:**<br>Myocardial ischemia: 11(61%)<br>Others: 7 (40%)*** |
| Bohm [62] | OHCA of presumed cardiac (Utstein 2004) | Exercise (n=147) | CAD: 25.8%<br>Cardiomyopathy:21%<br>Idiopathic VF:8%<br>Aortic dissection:1.1% |
| Weizman [63] | OHCA with not obvious cause (Utstein 2004) | Light to vigorous exercise(N=760) | **Women**:<br>Unknown:35%<br>Idiopathic: 25%<br>MI:33%<br>Dilated cardiomyopathy:7%<br>Hypertrophic cardiomyopathy: 2%<br>Electrical heart disease:5%<br>Non-cardiac etiologies:2%<br>**Men:**<br>Unknown:38%<br>Idiopathic: 28%<br>MI:30%<br>Dilated cardiomyopathy:3%<br>Hypertrophic cardiomyopathy: 1%<br>Electrical heart disease:3%<br>Non-cardiac etiologies:1% |

**AAD:** Ascending aorta dissection**. ACS**: Acute coronary syndrome. **CA**: Cardiac arrest. **CAD**: Coronary artery disease. **CI**: Confidence interval. **MET**: Metabolic equivalent. **MI**: Myocardial infarction. **OHCA**: Out-of-hospital cardiac arrest. **PE**: Pulmonary embolism. **STEMI**: ST-elevation myocardial infarction**. SAH:** Subarachnoid hemorrhage. **SR-OHCA**: Sport-related out-of-hospital cardiac arrest. **VF**: Ventricular fibrillation.

*Physical activity at the time of the incident was defined as one of following exercises: bicycling, conditioning exercises, dancing, fishing, hunting, sports, walking, running, water activities, and winter activities. OHCA occurring not during exercise defines as CA occurring during other activities, including home activity, inactivity, transportation, and occupation.

**Cardiac arrest occurred during exercise and marathon running.

***Other cardiac etiologies include coronary dissection, myocardial bridging, and anomalous connection of right coronary artery.

One study demonstrated that higher incidence of OHCA due to presumed cardiac was observed among an older cohort (>65 years old) with unhealthy level of pollutant standard index (PSI) than younger cohort (<65 years old) (RR:1.44 (1.29–1.69), p<0.001 vs 1.29 (1.06–1.58), p:0.012) [75]. This study identified risk of OHCA due to respiratory diseases was not associated with unhealthy levels of PSI (RR: 0.87(0.36–2.11, p=0.7) [75]. An interquartile increase in PM 2.5 was associated with higher risk of AMI (confirmed etiology) (Lag 0 hour: OR=1.14(1.03–1.27, Lag 1 hour: OR:1.14(1.03–1.26)) [85]. (Table 6)

Ten studies examined the association of ambient temperature with OHCA [44–46,50,53,54,57,58,86,87]. The incidence of OHCA due to presumed cardiac etiologies increased during cold and hot seasons. The risk of OHCA occurrence due to presumed cardiac etiologies was higher during cold or hot seasons than other seasons (<5°centigrade (C) (OR:1.20 (1.16–1.25)) or >30 °C (OR:1.11(1.04–1.18)) vs 5–9 °C (OR: 1.1(1.07–1.13))) [64], (winter: OR: 2.39 (1.3–4.3), p=0.045 vs spring:OR:0.91(0.43–1.8)) [65]. The incidence of presumed cardiac etiologies (Utstein 2004) of OHCA was higher during February than July (12% vs 6%:p<0.001)) [78] and during cold season compared with warm season (58.9% vs 53.9%) p<0.0001) whereas the incidence of OHCA due to respiratory disease was higher during warm seasons compared with cold season (6.2% vs 5.9%, p:0.036) [73]. (Table 6)

Further, the risk of OHCA of presumed cardiac etiologies was higher among older patients (> 65 years old) than younger (<65 years old) during heat wave (RR: 1.3 (95% CI 1.1–1.5) vs 1.1 (95 CI:0.9–1.1)) [70] and cold wave (RR: 1.451(1.039–2.028) vs 1.387(1.005–1.913), p<0.05) [86], (OR: 1.09 (1.02–1.16) vs (1.02 (0.93–1.11) [87]. Two studies suggested the risk of OHCA may increase by 7% (95% CI: 4−10%) [87] with any additional cold day (Lag 0−1 day: RR:1.12 (1.03–1.17); p<0.05; Lag 0−14 day:1.43 (1.14–1.78) [86].

Two case- crossover studies examined the association of wildfire smoke on the incidence of OHCA [81,82]. Dennekamp.et.al. reported the incidence of presumed cardiac (Utstein 2004) etiologies of OHCA increased by 24.6% (95% CI 4.5–48) and 5.4%(0.9–10.2) p<0.05) for an IQR (interquartile range) increase in CO and PM 2.5μ during fire season [81]. Another study demonstrated that the incidence of OHCA due to respiratory disease was significantly higher during wildfire period and among patients who were exposed to smoke in California compared with non-exposed patients (17% vs 6.9%) [82]. Probability of OHCA due to respiratory or presumed cardiac etiologies following smoke exposure was higher in lag day 2 (OR:1.7(1.18–2.13) and day 0 (OR:1.56(1.05–2.33) than lag day 1 (1.20 (0.8–1.79)) [82]. One time series study showed that the incidence of OHCA due to respiratory disease was higher during thunderstorms compared with the control time interval without thunderstorms (52.9% vs 0) [83]. (Table 6)

### Prodromal symptoms and prior healthcare utilization

Table 7 demonstrates nine studies, in total, that investigated the contribution of prodromal symptoms prior to OHCA to the etiology [88–90,92–97]. There was a variation in reporting of prodromal symptoms prior to OHCA for cardiac and non-cardiac etiologies. (S6 Appendix)

One case-crossover study investigated the healthcare consumption one week prior to OHCA and compared it with the same time one year before the occurrence of OHCA on the same population [90]. Chest pain was the predominant symptom to seek medical care one week prior to OHCA in comparison with control week (14% vs 0%), followed by gastrointestinal symptoms (7.7% vs 1.2%), and dyspnea (6.9% vs 0.2%) [90]. Chest pain was the main complaint to seek healthcare visits among patients with confirmed cardiac etiologies of OHCA when compared with other etiologies of OHCA (20.1% vs 2.5%) [90]. (Table 7)

### Discussion

This comprehensive scoping review focused on the circumstance-related factors of OHCA and their potential to contribute to the refinement of the etiology of OHCA. The main finding of this review is that ascertaining the etiology of cardiac arrest is complicated, and many pre-existing medical conditions or circumstance-related factors may directly or indirectly

**Table 6. Summary of articles evaluating the association of environmental factors with out-of-hospital cardiac arrest (OHCA) etiologies.**

| Author | Included OHCA population by etiology | Association of environmental stress with the increased risk of OHCA. % | Association of environmental factors with the risk of OHCA. Reported result in OR, or RR (if reported) |
|---|---|---|---|
| **Cold or hot ambient temperature** | | | |
| Yamazaki [64] | Presumed cardiac (Utstein 2004) | | **>30 °C**: OR:1.11(1.04–1.18)<br>**<5 °C:** OR:1.20 (1.16–1.25)<br>**5−9 °C:** OR:1.10 (1.07–1.13) |
| Yoshinaga [65] | Presumed cardiac (Utstein 2004) | | **<0 °C:(December, January, February)**:<br>OR: 1.52 (1.009–2.13) p = 0.045<br>**Winter**: OR: 2.39 (1.31–4.36; p = 0.004) |
| Tanigawa [66] | Presumed cardiac (Utstein 2004) | **<0 ° C**: 15%<br>**2-5°C decrease in the average temperature:**<br>Non-elderly:11%<br>Elderly:16% | **Every 5 °C increase in temperature:** OR: 0.89(0.86–0.91) |
| Kang [70] | Presumed cardiac (Utstein 2004) | **1 °C** increase in maximal temperature was associated with a 1.3% increases the risk of OHCA (p = 0.02) | **1 °C** increase in maximal temperature is associated with<br>>65 years: RR:1.3 (1.1–1.5) p = 0.03<br><65 years: RR: 1.1 (0.9–1.3) |
| Hensel [74] | Presumed cardiac (Utstein 2004) | **Probability of OHCA occurrence increased**:<br><5 °C: by 10% (95% CI 3–18%) p:0.011<br>5-25 °C: reference group<br>>25 °C: by 19% (95% CI 10–26%) p:0.028 | |
| Fukuda [73] | Presumed cardiac (Utstein 2004) | **OHCA occurred during**<br>Cold season: 6.4%<br>Midseason:66.9%<br>Warm season:26.7%<br>**OHCA due to cardiac cause**:<br>Cold season: 58.9%<br>Midseason: 56.4%<br>Warm season: 53.9%(p < 0.0001)<br>**OHCA due to respiratory disease**:<br>Cold:5.9%<br>Midseason:5.8%<br>Warm:6.2 (p:0.036) | |
| Nakanishi [78] | Presumed cardiac (Utstein 2004) | Seasonal variation:<br>Rate of OHCA: Feb (12%) vs July (6%)<br>in 3 coldest months: 71%<br>(95% CI: 45.2%−97.2%; p < 0.002)<br>71.2% more cardiac arrest occurred in 3 coldest months | |
| Nishiyama [77] | Presumed cardiac (Utstein 2004) | **The increased risk of OHCA with temperature <5 °C:**<br>Sleeping: 8.49%<br>Bathing: 111.42%<br>Working: 0.9%<br>Exercising: 3.84% | **OR of OHCA with 1 °C increase in temperature based on activity:**<br>Sleeping: OR:0.97(0.972–0.98)<br>Bathing: OR:0.91(0.90–0.923)<br>Working: OR:0.99(0.98–1.007)<br>Exercising: OR:1.004(0.97–1.03) |
| Dai [86] | Presumed cardiac (Utstein 2004) | | Cold ambient temperature<br>**Lag 0−1**: RR:1.102(1.03–1.17) p < 0.05<br><65 years old: RR: 1.101(1.005–1.207)<br>>65 years old: RR:1.098(1.003–1.203)<br>**Lag 0−14**: RR: 1.43(1.14–1.78) p < 0.05<br><65 years old: RR: 1.451(1.039–2.028)<br>>65 years old: RR: 1.387(1.005–1.913) p < 0.05 |
| Ryti [87] | Medical etiologies (Utstein 2015) | Additional cold temperature day:7% (4–10%) | **Winter**: OR:1.06(1.01–1.12)<br>**Autumn**: OR:1.06(1–1.12)<br>**Spring**: OR:1.08(1.02–1.14)<br>**Summer**: OR:1.07(1–1.15) |

*(Continued)*

| Author | Included OHCA population by etiology | Association of environmental stress with the increased risk of OHCA. % | Association of environmental factors with the risk of OHCA. Reported result in OR, or RR (if reported) |
|---|---|---|---|
| **Air pollution:PM2.5, PM10, Co, O3, and PNC** | | | |
| Wichmann [67] | Presumed cardiac (Utstein 2004) | **5 IQR increase in PM2.5:**<br>Lag day 2: 2.4% (95% CI −1.8–6.8%)<br>lag day 3: 4.4% (95% CI 0.2–8.8)<br>Lag day 4: 5.2(95% CI 1–9.5%)<br>**5 IQR increase in PM 10**<br>Lag day 2: 2.1% (95% CI −1.3–5.7)<br>Lag day 3:4.7% (95% CI 0.7–8.8)<br>Lag day 4:3.8(95% CI 0.2–7.6) | |
| Silverman [68] | Presumed cardiac (Utstein 2004) | | 10 mg/m3 increase in **PM 2.5**:<br>RR of CA:1.06(1.02–1.10)<br>Warm season: RR:1.09(1.03–1.15)<br>Cold season: RR:1.01(0.95–1.07) |
| Kang [69] | OHCA of presumed cardiac (Utstein 2004) | **10 µg/m3 increase in PM2.5**<br>1.3% (0.2–2.41%; p = 0.02)<br>**PM2.5 > 50µg/m3**:<br>13.4% increase in the risk of OHCA(p < 0.001) | **10 µg/m3 increase in PM2.5:**<br>Lag day1(ER:0.94% per 10 mg/m3; 95%CI −0.05–1.94; p = 0.062)<br>Lag day 2 (ER:1.13%;95%CI 0.16–2.11; p = 0.023) |
| Forastiere [72] | OHCA with presumed ischemic heart disease | **Risk of OHCA at Lag 0**:<br>PNC: 7.6% (95% CI 2–13.6%)<br>PM10: 4.8% (95%CI 0.1–9.8%)<br>CO: 6.5% (1–12.3%)<br>**lag 0–1:**<br>PNC:8.3% (1.8–15.2%)<br>PM10:6.1% (0.6–11.9%)<br>CO:7% (0.8–13.7%) | |
| Ho [75] | OHCA with all etiologies | | **Unhealthy level of pollutant standards index (PSI)**<br>**1-RR of OHCA**:1.37(1.2–1.56)<br>>=65 years: RR:1.44(1.22–1.69), p < 0.001<br><65 years: RR:1.29(1.06–1.58)<br>**2-Presumed cardiac etiologies**<br>Yes: RR:1.36(1.17–1.59) p < 0.001<br>No: RR:1.36(1.08–1.7)<br>**3- Respiratory disease**<br>Yes: RR: 0.87(0.36–2.11), p = 0.7<br>No: RR: 1.37(1.2–1.56) |
| Dennekamp [76] | OHCA with presumed cardiac etiologies (Utstein 2004) | **An IQR increase in PM 2.5 is associated with the risk of OHCA**<br>Lag 0: 2.44% (95%CI 0.54–4.37%)<br>lag 1:2.46%(95% CI 0.33–4.65%)<br>Average lag day 0 and 1:3.61% (95% CI 1.29–5.9%)<br>**The risk of OHCA**<br>Male: 4% (95%CI:1.18–6.9%)<br>Female: 2.77% (−1.28–6.99%)<br>65-74 yrs: 5.6% (0.33–11.15%)<br>>75yrs: 3.2% (−4.8–11.96) | |
| Kojima [71] | OHCA of presumed cardiac (Utstein 2004) | **10 µg/ m³ increase in PM2.5 is associated with** 1.6% increased risk of OHCA | |
| Levy [79] | OHCA of presumed cardiac etiologies (Utstein 2004) | | **An IQR increase in PM10 is associated with the risk of OHCA**<br>Lag0: RR: 0.89 (0.78–1.02)<br>Lag 3: RR: 1.01 (0.9–1.12)<br>**CO** (RR:0.99 (0.83–1.18)<br>**SO2** (RR:0.87(0.76–1) |

*(Continued)*

| Author | Included OHCA population by etiology | Association of environmental stress with the increased risk of OHCA. % | Association of environmental factors with the risk of OHCA. Reported result in OR, or RR (if reported) |
|---|---|---|---|
| Straney [80] | OHCA with presumed cardiac etiologies (Utstein 2004) | | **Per unit increase in PM 2.5**<br>Lag Day 0–8:OR:1.006(1–1.011) p<0.05<br>0-12:OR:1.007(1–1.013)<br>Lag hour 0–24:OR:1.009(1.002–1.016)<br>0-48:OR:1.01(1.002–1.018)<br>48hr average of PM2.5 was associated with a 13.6% increased risk of OHCA (OR:1.136, 95%CI 1.05–1.22)<br>**Per unit increase in CO:**<br>Lag day 0–8:OR:1.09(0.98–1.2) p<0.01<br>0-12:OR:1.114(1.01–1.22) p<0.05<br>Lag hour 0–24:OR:1.058(0.93–1.19)<br>0-48:OR:1.04(0.90–1.19) |
| Gentile [84] | OHCA with medical etiologies (Utstein 2015) | | Benzene: OR:2.3 (1.6–2.7) p<0.001<br>PM 10: OR:1.01(1–1.02) p=0.01<br>PM 2.5: OR:1.02(1–1.04) p<0.002<br>CO: OR:10.6(3.3–36.5) p<0.001 |
| Rosental [85] | OHCA with confirmed etiologies of AMI and other cardiac etiologies | | **An IQR increase in PM2.5 level is associated with risk of**<br>**AMI:**<br>Lag hour 0: OR:1.17(1.03–1.33)<br>Lag day 0: OR:1.14(1.03–1.27)<br>Lag 1d: OR:1.14(1.03–1.26)<br>Lag 2d: OR:1.11 (1–1.23)<br>**An IQR increase in O_3 level is associated with other cardiac etiologies of OHCA:**<br>Lag hour 0: OR:1.02(0.9–1.15)<br>Lag0d: OR:1.16(0.99–1.36)<br>Lag 1d: OR:1.26(1.07–1.48)<br>Lag2d: OR:1.3(1.11–1.53) |
| **Other environmental conditions** | | | |
| Dennekamp [81] | OHCA with presumed cardiac etiologies | **An IQR increase in air pollutant (PM 2.5):**<br>**Fire smoke Season**<br>Lag 0: 1.9% (95% CI-0.6–4.5)<br>Lag hour 0–24:3.5% (95% CI −0.1–7.3)<br>Lag hour 0–48: 5.4% (95% CI 0.9–10.2), p<0.01<br>**Whole year:**<br>Lag 0: 1.3% (95%CI −1–3.8)<br>Lag hour 0–24: 3%(95% CI −0.3–6.5)<br>Lag hour 0–48:4.4% (95% CI 0.2–8.7%), p<0.01<br>**an IQR increase in CO:**<br>**Fire smoke season:**<br>Lag 0:3.9%(−6–14.8)<br>Lag hour 0–24:16.5% (−0.1–35.8)<br>Lag hour 0–48:24.6% (4.5–48) p<0.05<br>**Whole year:**<br>Lag 0: 0.6% (95% CI −4.8–3.9%)<br>Lag hour 0–24: 2.7%(95% CI −3.3–9.2%)<br>Lag hour 0–48-: 5.6%(95% CI −1.6–13.2%) | |
| Andrew [83] | OHCA with confirmed etiologies | **Thunderstorm period:**<br>Cardiac: 35.3%<br>Respiratory: 52.9%<br>**Comparator period:**<br>Cardiac: 60%<br>Respiratory: 0 | |

*(Continued)*

**Table 6.** (Continued)

| Author | Included OHCA population by etiology | Association of environmental stress with the increased risk of OHCA. % | Association of environmental factors with the risk of OHCA. Reported result in OR, or RR (if reported) |
|---|---|---|---|
| Jones [82] | OHCA with presumed cardiac etiologies | **Wildfire smoke**<br>**Exposed patients:**<br>Presumed cardiac:811(16.3%)<br>Respiratory disease: 66(17.9%) | |
| | | **Non-exposed patients:**<br>Presumed cardiac: 4967(93%)<br>Respiratory:369(6.9%) | |

**AMI:** Acute myocardial infarction. **C**: Centigrade. **CO**: Carbon monoxide. **ER**: Excess ratio. **IQR**: Interquartile range. **O₃**: Ozone. **OHCA**: Out-of-hospital cardiac arrest. **OR**: Odd ratio. **PNC**: Particle number concentration. **PSI**: Pollutant standard index. **PM**: Particle matter. **RR**: Risk ratio. **SO₂**: Sulfurous oxide.

contribute to the etiology of cardiac arrest. This scoping review suggests that OHCA attributed to cardiac or obvious non-cardiac etiologies may be more accurately precipitated by psychiatric disease, antipsychotic or antidepressant medications, MI, seizure, concurrent Covid-19 infection, alcohol intoxication, illicit or recreational drug use, and suicide (Table 3). Identifying potential contributing factors relies on diligent initial assessment of patients' signs and symptoms, medical history, drug paraphernalia, and prescribed or non-prescribed medications. Our results support a prior recommendation that clearly defined case definitions for each etiology and contributing factors may enable prehospital personnel and data abstractors to identify more likely etiology of OHCA and report the etiology more consistently [14]. These identified contributing factors may also explain the previous observed discrepancy between initial and confirmed etiologies of OHCA [5] because ascertaining the etiology of OHCA might be partially relied on prehospital personnel's ability to interpret patient's signs, symptoms, and contributing factors along with circumstance-related factors.

There is limited evidence on contributing factors and etiology of OHCA. Our review identified the association of antipsychotic medications with initial presenting rhythm of PEA [34,35] which may be inconsistent with previous studies that suggested an association of antipsychotic or antidepressant drugs with QT interval prolongation and arrhythmia induced cardiac arrest (e.g., torsade de pointe) [98,99]. Further, distinguishing these medications as contributing from causative factors may depend on the blood level of these medications (toxic level vs therapeutic level) or daily dose of medication (low vs high dose) [13,98,99], but in our review, just three included studies investigated the blood level of medications (e.g., opioid, recreational, and prescribed medications) in non-survivors of OHCA [7,13,41]. Blood levels of medications may be identified through reviewing medical charts, toxicology tests, or autopsy reports which highlights the importance of data linkage, reviewing multiple sources, and comparing initial and confirmed etiologies of OHCA to identify the primary and secondary etiologies of OHCA and improve consistency in reporting incidence and outcomes across all studies. Although our review couldn't identify any clear evidence of drug paraphernalia [100], one study suggested an algorithm in the case of absence of drug paraphernalia or witness to help prehospital personnel consider the probability of drug overdose and naloxone administration in the field, which might improve the outcome of OHCA. According to this algorithm, patients who are younger than 40, female, and black or white race were more probably (40%) suspected of OD-OHCA than male, older than 60 years old, witnessed OHCA, and other races (e.g., Asian and Latinos) [41].

Factors related to the circumstance and context, including exercise or environmental factors might trigger OHCA and may contribute to refining etiological classification. A systematic review on the association of triggers with fatal cardiac arrest suggested annual sport-related cardiac death was rare at <2 per 100,000 athletes [101]. Our review identified nine studies with reported annual incidence of SR-OHCA of 0.01–4.7 per 100,000 athletes which is comparable. This scoping review suggests that exercise triggers cardiac arrest predominantly due to cardiac etiologies and infrequently attributed

**Table 7. Summary of studies evaluating the association of reported prodromal symptoms with the etiology of out-of-hospital cardiac arrest (OHCA).**

| Author | Presumed initial etiology: N | Final etiology;N (%) If reported | Reported prodromal symptoms: N(%) | Association of reported prodromal symptoms and etiology; N (%) |
|---|---|---|---|---|
| Inamasu [88] | Presumed cardiac etiology (n=250) | Cardiac:225 SAH:8 PN:4 PE:6 AA/AD:7 | LOC:138 (55.2%) CP:70 (28%) | **LOC and diagnosed with non-cardiac causes**: SAH=8 PN=4 PE=4 AA/AD=2 (13% diagnosed with non-cardiac etiologies) **CP and diagnosed with non-cardiac etiologies:** AA/AD=5 PE=2 (10% diagnosed with non-cardiac etiologies) |
| Lee [89] | Cardiac etiologies (n=9,361) | | No prodromal symptoms: 59.5% Neurological symptoms: 14.2%* Respiratory symptoms: 12.5% Cardiac symptoms: 5% GI symptoms: 5% | **Cardiac etiologies:** No symptom:5,839(62.4%) Cardiac symptoms:468(5%) Respiratory symptoms:1171(12.5%) Neurologic symptoms: 1328(14.2%) GI symptoms: 466 |
| | Non-cardiac (n=3,424) Other etiologies (n=184) | | | **Non-cardiac etiologies:** No prodromal symptoms: 1773(51%) Cardiac symptoms: 30(0.9%) Respiratory symptoms: 221(6.5%) Neurologic symptoms: 633(18%); GI symptoms: 68(2%) |
| Hoglung [90] | Presumed cardiac etiology(n=403) | Definitive or probable MI: 279 (69%) | **One-week prior to CA:** Chest pain: 59(14.6%) GI symptoms: 31(7.7%) Dyspnea: 28(6.9%) | **The prevalence of chest pain one week before MI**:20.1% |
| | | | **Same period one year before CA:** Chest pain: 0 (p<0.001) GI symptoms:5(1.2%) Dyspnoea:1(0.2%) | **The prevalence of chest pain one week before other etiologies**: 2.5% |
| Nehme [97] | Presumed cardiac etiologies (n=1,056) | Not reported | Chest pain:48.8%** Dyspnea: 41.8% Altered consciousness: 37.8% | |
| Kurckiyan [92] | PE(n=60) | PE:60 | Dyspnea:41(68%) Syncope:29(48%) Chest pain:15(25%) | |
| Arnaout [93] | Neurological etiologies (n=86) | SAH:73(85%) Subdural hematoma:(3%) Ischemic stroke:6% Intracerebral hematoma:3% | Neurological symptoms Non-neurological symptoms | **Neurological symptoms: 31(36%)**** Headache:17(20%), p<0.001 Impaired consciousness: 8(9%) Seizure:7(8%), p=0.03 Neurological deficit (focal signs):3(4%) **Other prodromal symptoms:**6(7%), p<0.001 Chest pain:3(4%), p:0.01 Dyspnea: 2(2%), p:0.001 Syncope: 1(1%), p:1 |
| | Non-neurological disease (n=172) | Cardiac:98 Respiratory and other etiologies:74 | | **Neurological Symptoms:** 14(8%) p<0.001 Headache: 2(1%) p<0.001 Impaired consciousness: 7(4%) Seizure: 4(2%) p=0.03 Neurological deficit (focal signs): 1(1%) **Other prodromal symptoms:**51(30%) Chest pain: 24(14%) Dyspnea: 30 (17%) Syncope:1(1%) |

*(Continued)*

**Table 7.** (Continued)

| Author | Presumed initial etiology: N | Final etiology;N (%) If reported | Reported prodromal symptoms: N(%) | Association of reported prodromal symptoms and etiology; N (%) |
|---|---|---|---|---|
| Nazerian [96] | All OHCA patients (n = 280) | **ABI: 21(7.5%)** Cerebral hemorrhage Spinal cord injury Ischemic stroke Status epilepticus | Neurological symptoms*** Non-neurological | **ABI:** Neurological prodromal symptoms: 12 (40%) Chest pain:0 Dyspnea:0 |
| | | **Cardiac disease** (ACS, Cardiogenic shock): 159(56.8%) **Hypoxemia** (Pneumonia, COPD): 27(9.6%) | | **Cardiac etiologies**: Chest pain: 65(25.1%) Dyspnea: 50(19.3%) |
| Marijon [94] | Presumed cardiac (n = 839) | | **No symptoms:** 409 (48%) **Presence of symptoms**: 430(52%) Chest pain: 199(46%) Dyspnea:78(18.1%) Syncope/ palpitation:22(5.6%) Others:127(29.5%) | **CAD related OHCA:** **No symptoms:** 218 (79%) **Presence of symptoms**: 240 (82%), p=0.16 |
| Nishiyama [95] | Presumed Cardiac (n = 1,042) | **Cardiac**:1042 | Prodromal symptoms 61.8%, (p=0.003) | **Cardiac etiologies:** Chest pain: 20.7% Dyspnea: 27.6% Syncope:12% Cold sweet: 3.4% |
| | Non-cardiac (n = 424) | **Non-Cardiac**: 424 Cerebrovascular disease:67(15%) Respiratory disease:92(21%) Aortic disease:44(10%) Malignancy:93(21%) | Prodromal symptoms:70% | **Non-cardiac etiologies:**** Chest pain:3.4% Dyspnea:40.7% Syncope:14.5% Cold Sweet: 1.7% |

**AA:** Aortic aneurysm. **ABI**: Acute brain injury. **ACS**: Acute coronary syndrome. **AD**: Ascending aortic dissection**. CAD:** Coronary artery disease. **COPD**: Chronic obstructive pulmonary disease. **CP**: Chest pain. **GI**: Gastrointestinal. **LOC**: Loss of consciousness. **MI**: Myocardial infarction. **OHCA**: Out-of-hospital cardiac arrest. **PE**: Pulmonary embolism. **PN**: Pneumonitis. **SAH**: Subarachnoid hemorrhage.

*Neurological symptoms include mental change, seizure, headache, convulsion, dizziness, and paralysis.

**The association of prodromal symptoms with final underlying etiologies not reported.

***Neurological symptoms: seizure, headache, and focal signs.

to non-cardiac etiologies. A slight variation was observed in case definition, source of extracting data (e.g., media, FIFA website), and the underlying etiology of cardiac arrest across included studies. Etiology of OHCA may vary between competitive and non-competitive athletes or between vigorous (MET > 6) and light (MET 0–3) exercise intensity. This finding might be in line with other studies that coronary artery disease and ST elevation myocardial infarction (STEMI) were the predominant etiologies of non-exercise or low intensity (MET < 6) exercise- related cardiac death (67 vs 14%) [102] [103]. However, cardiomyopathy, inheritable structural cardiac disease, and arrhythmia more likely contribute to exercise- related cardiac arrest in younger athletes (<35 years old) [103,104]. These findings suggest that integrating exercise-related data into the Utstein reporting template, including a standardized case definition of exercise-induced OHCA and intensity of exercise across all data registries may optimize etiology ascertainment.

Environmental triggers are another factor that may contribute to cardiac arrest. Different air pollutants (PM2.5, CO, and $O_3$) and cold or hot ambient temperature are all associated with presumed cardiac etiologies of OHCA while smoke

from wildfires might trigger respiratory diseases which may consequently result in cardiac arrest. These associations are hypothetical since most of the included studies did not confirm the etiology of cardiac arrest. This finding may be consistent with another study that reported an increase in PM 2.5 was associated with higher hospital admission and incidence of MI (without experiencing cardiac arrest) (OR:2.92 (95%CI: 2.22–3.83; P:0.001) [105]. A systematic review suggested cold ambient temperature was associated with higher mortality rate of cardiovascular disease (RR:1.11 (1.03–1.19) and respiratory diseases (RR: 1.21 (0.97–1.51) [19], but our scoping review identified slightly inconsistent results from just one study that reported the incidence of OHCA due to respiratory disease increased in warm seasons [73]. As the included population in our review may be very etiologically heterogeneous, consistent data validation may help identify the association of these environmental triggers with confirmed etiologies of OHCA. These triggers may potentially contribute to the explanation of the observed variability in reported incidence of presumed cardiac etiologies of OHCA across data registries [106].

The last finding of this review was that approximately half of patients with OHCA experienced prodromal symptoms before cardiac arrest attributed to cardiac and non-cardiac etiologies which may strengthen the previous systematic review of 911 calls suggested that 70% of patients with OHCA reported breathing problems prior to cardiac arrest; however, the etiology was not confirmed [107].This scoping review further identified different underlying etiologies of OHCA may have distinct presentations, which may need different treatment and care. Non-neurological etiologies were more likely associated with prodromal symptoms of chest pain (14% vs 4%) and dyspnea (17% vs 2%) when compared with neurological etiologies [93]. Dyspnea was more likely associated with non-cardiac etiologies (e.g., PE), and chest pain was the hallmark of cardiac etiologies of OHCA. There is considerable overlap in symptoms and etiology. This review also identified that 20% of patients with confirmed cardiac etiologies of OHCA sought medical care for chest pain within one week prior to cardiac arrest. This finding might strengthen another study's result that 70% of patients with OHCA (in general) regardless of the etiology of cardiac arrest used the healthcare system within 90 days prior to cardiac arrest [108]. Hence, prodromal symptoms may contribute to assigning the more likely etiology of cardiac arrest or even seeking healthcare visit prior to cardiac arrest is an important contributing factor.

## Limitations

This scoping review has some limitations that should be considered when interpreting the results. Potential selection bias exists for this scoping review due to language restrictions to English and the lack of inclusion of gray literature. Just one reviewer screened all studies due to the infeasibility of having the second reviewer with content expertise and restricted time. All the included studies were predominantly published in developed countries with data from resuscitation registries. No study has been found from developing countries which might be the result of restricting the search to English language. We limited our review to OHCA as the focus was on defining out of hospital contributing factors, triggers, or prodromal symptoms that may contribute to more accurate etiological classification of OHCA. Over 50% of actively treated OHCA currently die in the field and better etiological classification in the field may reveal treatment options and affect reporting outcomes.This scoping review was not aiming to evaluate the accuracy of etiological classification of OHCA; however, the etiological misclassification is potential. This scoping review provides an overview of the breadth of current published articles and should not be interpreted as a quality appraisal of current evidence.

## Conclusion

Ascertaining the underlying etiology of OHCA is challenging, but pre-existing medical conditions and circumstance-related factors contribute to cardiac arrest, predominantly due to cardiac etiologies. Vigorous initial assessment of pre-existing medical conditions, signs, and symptoms, along with circumstance-related factors may enable prehospital personnel and data abstractors to ascertain the primary and secondary etiologies of cardiac arrest. Including these factors in the Utstein reporting template refines the body of evidence and quality of future studies which may lead to better understanding of

 

the association of these circumstance-related factors with the etiology of OHCA. Better OHCA reporting by etiology will contribute to uniform reporting of OHCA across all data registries.

## Supporting information

**S1 Appendix.  Preferred Reporting Items for Systematic reviews and Meta-Analyses extension for Scoping Reviews (PRISMA-ScR) Checklist.**
(DOCX)

**S2 Appendix.  Search strategy for conducting this scoping review.**
(DOCX)

**S3 Appendix.  Summary of included studies evaluating the association of contributing factors with out-of-hospital cardiac arrest (OHCA) etiologies.**
(DOCX)

**S4 Appendix.  Summary of included studies evaluating etiologies of exercise-related out-of-hospital cardiac arrest (OHCA).**
(DOCX)

**S5 Appendix.  Summary of included studies focused on the association of environmental factors with out-of-hospital cardiac arrest (OHCA) etiologies.**
(DOCX)

**S6 Appendix.  Summary of included studies evaluating the association of prodromal symptoms with out-of-hospital cardiac arrest (OHCA) etiologies.**
(DOCX)

## Acknowledgments

The authors would like to thank David Lightfoot, the information specialist at St. Michael's Hospital and librarians at the University of Toronto for their contribution on developing search strategies and searching articles through electronic resources.

## Author contributions

**Conceptualization:** Sedigheh Shaeri, Julie Considine, Theresa Mariero Olasveengen, Laurie J. Morrison.

**Data curation:** Sedigheh Shaeri.

**Formal analysis:** Sedigheh Shaeri.

**Investigation:** Sedigheh Shaeri.

**Methodology:** Sedigheh Shaeri, Katie N. Dainty, Laurie J. Morrison.

**Project administration:** Sedigheh Shaeri, Laurie J. Morrison.

**Supervision:** Laurie J. Morrison.

**Visualization:** Sedigheh Shaeri.

**Writing – original draft:** Sedigheh Shaeri.

**Writing – review & editing:** Sedigheh Shaeri, Julie Considine, Katie N. Dainty, Theresa Mariero Olasveengen, Laurie J. Morrison.

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
