## [Decision Letter · Decision Letter 0]

Thank you for submitting your manuscript to PLOS ONE. After careful consideration, we feel that it has merit but does not fully meet PLOS ONE’s publication criteria as it currently stands. Therefore, we invite you to submit a revised version of the manuscript that addresses the points raised during the review process.

We look forward to receiving your revised manuscript.

Kind regards,

Nik Hisamuddin Nik Ab. Rahman

Academic Editor

PLOS ONE

 [The primary author received 2019-2020 Graduate student funding from Institute of Health Policy, Management, and Evaluation and Graduate Student Endowment Funding from faculty of medicine, University of Toronto.]. 

Reviewers' comments:

Reviewer's Responses to Questions

**Comments to the Author**

1. Is the manuscript technically sound, and do the data support the conclusions?

Reviewer #1: Yes

Reviewer #2: Yes

2. Has the statistical analysis been performed appropriately and rigorously?

Reviewer #1: Yes

Reviewer #2: Yes

3. Have the authors made all data underlying the findings in their manuscript fully available?

Reviewer #1: Yes

Reviewer #2: Yes

4. Is the manuscript presented in an intelligible fashion and written in standard English?

Reviewer #1: Yes

Reviewer #2: Yes

Reviewer #1: The manuscript is organized, methodologically sound, and relevant to emergency medicine and cardiac arrest classification.

Overall Assessment of the Manuscript

This manuscript presents a well-organized and methodologically robust scoping review of the factors, triggers, and early symptoms contributing to the etiological classification of out-of-hospital cardiac arrest (OHCA). The research is timely and essential, addressing a vital gap in understanding OHCA etiology, with significant implications for clinical practice and future research. Below is a comprehensive critique that evaluates the content, clarity, methodology, and adherence to PRISMA-ScR guidelines.

Areas for Improvement

1. Justification for Scoping Review Approach (Minor Issue)

Though the study utilizes a scoping review methodology, the rationale for selecting this approach could be articulated more effectively. Incorporate a brief statement in the introduction that clarifies why a scoping review (as opposed to a systematic review) was the most suitable choice.

2. Transparency in Protocol Registration (Moderate Issue)

The manuscript notes that the protocol is registered with the Open Science Framework (OSF) but does not include a registration number or direct link. Add the OSF link for increased transparency.

3. Missing Detailed Search Strategy (Moderate Issue)

The search strategy is summarized but lacks comprehensive detail, such as Boolean operators, MeSH terms, and gray literature inclusion. Please Provide a complete search strategy in an accompanying appendix.

4. Justification for Excluding Critical Appraisal (Moderate Issue)

The manuscript does not explain why a formal quality assessment is excluded. Although PRISMA-ScR does not require such an assessment, a brief justification is essential. please Include a one-sentence rationale in the methods section.

5. Limitations Section Needs Expansion (Major Issue)

The limitations section is brief and does not discuss potential biases (e.g., selection bias, exclusion of non-English studies, reliance on EMS data). Please expand the limitations to include:

◦ Potential selection bias due to language restrictions.

◦ Inherent limitations of EMS data (e.g., misclassification of OHCA etiology).

◦ Absence of a formal quality appraisal as a potential limitation.

◦ Future research is necessary to validate findings with prospective cohort studies.

Reviewer #2: Thank you for the opportunity to review this well-conducted scoping review.

This is a systematic review aimed at identifying additional contributing factors, triggers, and symptoms of out-of-hospital cardiac arrest (OHCA) to supplement the existing Utstein criteria with further classification parameters. Overall, this systematic review has been conducted with sound scientific methodology.

However, it appears that a substantial number of studies have already examined the relationship between COVID-19 and OHCA. I find it difficult to understand why COVID-19 is considered such a significant influencing factor in OHCA, and I would appreciate clarification from the authors on this matter. The included studies in this systematic review seem to be heavily focused on COVID-19, which I would like to comment on and critically question.

Apart from this, the paper is coherently written, and this work represents a valuable contribution to the scientific understanding of OHCA and the development of more suitable quick alerts for identifying its etiology.

**Do you want your identity to be public for this peer review?** For information about this choice, including consent withdrawal, please see our Privacy Policy

Reviewer #1: **Yes: ** Assistant Professor of Nursing, Department of Prehospital Emergency Medicine, Asadabad School of Medical Sciences, Asadabad, Iran. Email: A.khazaei91@gmail.com, ORCID ID: 0000-0002-8063-3419

Reviewer #2: **Yes: ** Joachim Riße

---

## [Author Response · Author response to Decision Letter 1]

6 Jun 2025

Journal requirements and reviewers’ comments

Journal requirements

1-Thank you for stating the following financial disclosure:

[The primary author received 2019-2020 Graduate student funding from Institute of Health Policy, Management, and Evaluation and Graduate Student Endowment Funding from faculty of medicine, University of Toronto.].

The authors’ response: We declare that “The funders had no role in study design, data collection and analysis, decision to publish, or preparation of this manuscript”

2- "We note that your Data Availability Statement is currently as follows: [All relevant data are within the manuscript and its Supporting Information files.]

Authors' response: This is a scoping review to map current published articles. No meta-analysis has been conducted. Minimal data set of this scoping review, including a summary of included studies is available within the manuscript and S3-6 Appendices with relevant citations. The result of descriptive analysis (number and percentage) is available in Table 2 . No mean and standard deviation have been measured. There is no graph in this submission. No points have been extracted from images for analysis.

Reviewer #1

The manuscript is organized, methodologically sound, and relevant to emergency medicine and cardiac arrest classification.

Overall Assessment of the Manuscript:

This manuscript presents a well-organized and methodologically robust scoping review of the factors, triggers, and early symptoms contributing to the etiological classification of out-of-hospital cardiac arrest (OHCA). The research is timely and essential, addressing a vital gap in understanding OHCA etiology, with significant implications for clinical practice and future research. Below is a comprehensive critique that evaluates the content, clarity, methodology, and adherence to PRISMA-ScR guidelines. Areas for Improvement

1. Justification for Scoping Review Approach (Minor Issue) Though the study utilizes a scoping review methodology, the rationale for selecting this approach could be articulated more effectively. Incorporate a brief statement in the introduction that clarifies why a scoping review (as opposed to a systematic review) was the most suitable choice

Authors' response: The justification for scoping review is included in the introduction (Line# 85-87). We selected the scoping review because it was anticipated studies with diverse methodology (mostly observational or reflective) and limited number of publications. The purpose of this review was to map current evidence to identify the contributing factors, triggers, and prodromal symptoms of OHCA that may contribute to etiological classification of cardiac arrest. Since no systematic or scoping review has been conducted before, the scoping review was deemed to be the best option to map evidence and identify scientific evidence to address the study’s objective.

2. Transparency in Protocol Registration (Moderate Issue)

The manuscript notes that the protocol is registered with the Open Science Framework (OSF) but does not include a registration number or direct link. Add the OSF link for increased transparency.

Authors' response: The OSF provides DOI, which is specific to this scoping review and directly links to the registration (Line # 100 and S1 Appendix). The registration number OSF.IO/K5ZDP was added to line #99.

3. Missing Detailed Search Strategy (Moderate Issue)

The search strategy is summarized but lacks comprehensive detail, such as Boolean operators, MeSH terms, and gray literature inclusion. Please Provide a complete search strategy in an accompanying appendix.

Authors' response: Search strategy is available in S2 Appendix, including the chosen mesh terms and Boolean to search all three electronic databases, including Medline, Embase, and EBM-Cochrane databases.

We didn’t search gray literature, and this was noted as limitation of this scoping review (line#401). According to scoping review methodology, searching gray literature is not mandatory to conduct a scoping review [1]. We strongly preferred to map scientific evidence from published peer-reviewed studies for this scoping review to provide a rational knowledge synthesis and a firm foundation upon which to justify further research.

4. Justification for Excluding Critical Appraisal (Moderate Issue)

The manuscript does not explain why a formal quality assessment is excluded. Although PRISMA-ScR does not require such an assessment, a brief justification is essential. please Include a one-sentence rationale in the methods section.

Authors' response: Critical appraisal is not mandatory for conducting a scoping review because the purpose of scoping review is to provide a comprehensive evidence map or an overview of current evidence regardless of the quality of evidence [1-3]. This was explained and cited in Line # 142. One sentence has been added to the limitation. (line#416-417)

5. Limitations Section Needs Expansion (Major Issue)

The limitations section is brief and does not discuss potential biases (e.g., selection bias, exclusion of non-English studies, reliance on EMS data). Please expand the limitations to include:◦ Potential selection bias due to language restrictions.◦ Inherent limitations of EMS data (e.g., misclassification of OHCA etiology).◦ Absence of a formal quality appraisal as a potential limitation.◦ Future research is necessary to validate findings with prospective cohort studies. Thanks for comments, suggestions, and encouragement

Authors' response: We excluded the non-English literature according to our protocol and defined eligibility criteria (Table 1; Page#7) due to difficulty of translation or having someone with relevant expertise to translate the scientific text in foreign languages. This was already noted as the limitation of this study (Line# 401-402) and slightly revised.

We added one sentence to the limitation (line # 415-416) pertaining to the potential etiological misclassification of included studies. The purpose of this scoping review was not to evaluate the etiological misclassification. A prior scoping review highlighted a three-time discrepancy between initial etiological classification of OHCA based on EMS documentation using Utstein etiological classification and confirmed etiologies documented in medical charts or autopsy reports [4].This current scoping review was conducted to understand the role of contributing factors, triggers, and prodromal symptoms, which may help refine the etiological classifications. The finding of this review may strengthen the finding of previous scoping review on the potential etiological misclassification.

We included studies with observational, case-crossover, and time series that provide scientific evidence regarding the contributing factors, triggers, and prodromal symptoms of OHCA. The design of selected studies is available in S3-S6 Appendices. Future research would be suggested to integrate findings of this scoping review with real-life data. (Line#424)

Reviewer #2

Thank you for the opportunity to review this well-conducted scoping review. This is a systematic review aimed at identifying additional contributing factors, triggers, and symptoms of out-of-hospital cardiac arrest (OHCA) to supplement the existing Utstein criteria with further classification parameters. Overall, this systematic review has been conducted with sound scientific methodology.

However, it appears that a substantial number of studies have already examined the relationship between COVID-19 and OHCA. I find it difficult to understand why COVID-19 is considered such a significant influencing factor in OHCA, and I would appreciate clarification from the authors on this matter. The included studies in this systematic review seem to be heavily focused on COVID-19, which I would like to comment on and critically question.

Apart from this, the paper is coherently written, and this work represents a valuable contribution to the scientific understanding of OHCA and the development of more suitable quick alerts for identifying its etiology.

Thank you for your supportive comments.

Authors' response: Observational studies of COVID-19 pandemic and etiological classification of OHCA were targeted in the search as an example of a circumstance or contributing factor affecting the etiological classification of OHCA. The covid-19 pandemic as a contributing factor increased the likelihood of respiratory or medical cause to OHCA. One sentence has been added to clarify this section. (Line# 177)

---

## [Editor Report · Decision Letter 1]

The role of contributing factors, triggers, and prodromal symptoms in the etiological classification of out-of-hospital cardiac arrest; A Scoping Review

PONE-D-25-01818R1

Dear SEDIGHEH SHAERI,

We’re pleased to inform you that your manuscript has been judged scientifically suitable for publication and will be formally accepted for publication once it meets all outstanding technical requirements.

Kind regards,

Nik Hisamuddin Nik Ab. Rahman

Academic Editor

PLOS ONE
---

## [Editor Report · Acceptance letter]

PONE-D-25-01818R1

PLOS ONE

Dear Dr. SHAERI,

I'm pleased to inform you that your manuscript has been deemed suitable for publication in PLOS ONE. Congratulations! Your manuscript is now being handed over to our production team.

Kind regards,

on behalf of

Professor Dr Nik Hisamuddin Nik Ab. Rahman

Academic Editor

PLOS ONE